# Predicting Athlete Workload in Women’s Rugby Sevens Using GNSS Sensor Data, Contact Count and Mass

**DOI:** 10.3390/s24206699

**Published:** 2024-10-18

**Authors:** Amarah Epp-Stobbe, Ming-Chang Tsai, Marc D. Klimstra

**Affiliations:** 1Canadian Sport Institute Pacific, Victoria, BC V9E 2C5, Canadamtsai@csipacific.ca (M.-C.T.); 2School of Exercise Science, Physical and Health Education, University of Victoria, Victoria, BC V8P 5C2, Canada

**Keywords:** GNSS, load monitoring, contact sport, women’s sport

## Abstract

The use of session rating of perceived exertion (sRPE) as a measure of workload is a popular athlete load monitoring tool. However, the nature of sRPE means the contribution of salient, sport-specific factors to athlete load in field sports is challenging to isolate and quantify. In rugby sevens, drivers of load include high-speed running and physical contact. In soccer and men’s rugby, union acceleration/deceleration also influences load. These metrics are evaluated using data from global navigation satellite system (GNSS) sensors worn by athletes. Research suggests that sensor data methods for identifying load in men’s rugby do not accurately quantify female athlete loads. This investigation examined how mass, contact, and accelerations and decelerations at different speeds contribute to load in women’s rugby sevens. The study evaluated 99 international matches, using data from 19 full-time athletes. GNSS measures, RPE, athlete mass, and contact count were evaluated using a linear mixed-model regression. The model demonstrated significant effects for low decelerations at low and high speeds, mass, distance, and contact count explaining 48.7% of the global variance of sRPE. The use of acceleration/deceleration and speed from GNSS sensors alongside mass, as well as contact count, presents a novel approach to quantifying load.

## 1. Introduction

### 1.1. Monitoring Athlete Workload in Sport

Understanding the loads that athletes experience in competition may provide important benchmarks for optimizing preparedness [1,2]. Monitoring athlete load may be accomplished using various tools including athlete-worn sensors, video-based event tagging and ratings of subjective effort. While there is no gold-standard method applicable across sports, a common current approach is to use session rating of perceived exertion (sRPE) for both male and female team sports [3]. sRPE is the product of an athlete’s subjective self-reported rating of perceived exertion (RPE) and the objective duration of the session in minutes [3,4].

RPE is considered a reflection of an “internal” training load, as it reflects the athlete’s own perceived effort, rooted in a subjective interpretation of their performance relative to their maximal effort [3]. It has been found that while RPE alone does not entirely reflect the load experienced by an athlete, multiplying this subjective measure by distance or duration, as is achieved when calculating sRPE, provides a valuable metric for load determination [5,6,7]. sRPE has been shown to have relationships to other internal training load measures, like training impulse (TRIMP), including Bannister’s, Lucia’s, and Edwards’, as well as external load measures like total and high-speed distance [3].

While sRPE is popular across sports due to its ease of collection and simplicity in analyses, its subjective component (RPE) means that the influence of individual factors and their specific impact on athlete load are not clearly identified when using this metric alone [3,8]. The calculation of sRPE relies on self-reporting from athletes. Unfortunately, this subjective information is often missing at random, and without a consistent determination of load, it is difficult to monitor athlete performance properly [9,10,11]. Therefore, there is a need to identify specific objective factors that contribute to athlete load, and there exists a great opportunity to use a specialized kinematic analysis of athlete-worn sensor data to provide suitable metrics. Identifying and quantifying salient, objective, sport-specific drivers of athlete load should enable a more targeted approach to load monitoring to optimize performance and minimize injury risk [2,10,11].

This information will also be critical for sport coaches and practitioners to apply load metrics to objectively evaluate technical and tactical interventions [2,10,11]. Fortunately, the use of global navigation satellite system (GNSS) data collected from athlete-worn sensors during training and competition in team sports is of great value to objectively quantify an athlete’s performance.

### 1.2. Athlete Workload in Rugby Sevens–GNSS Metrics

As an international governing body, World Rugby mandates the use of specific monitoring tools, such as particular GNSS units, to be worn in competition [12]. GNSS sensors provide speed and distance data at 10–20 Hz and are often used to provide average or cumulative kinematic variables, such as providing an overall distance or duration of activity for use in sRPE calculations. While these variables are important performance summaries, this simple approach to GNSS sensor data analysis does not provide adequate detail of the unique demands experienced by the athlete, such as those during bouts of high-intensity effort. This is of particular importance to note, as high-speed running events and physical contact have been identified as two of the many objective drivers of load in rugby [12,13,14]. Therefore, there have been efforts to further identify more detailed objective kinematic measures available through GNSS, such as the quantification of bouts of high-speed running, and compare these metrics to the established athlete load measure (sRPE) [3,7,15,16,17]. From these investigations, it has been shown in men’s rugby research that there are connections between high-speed running, distances covered and athlete load (sRPE) [3,7]. Delaney et al. (2018) demonstrated that accelerations and decelerations are drivers of load in men’s rugby leagues, as well as show associations to metabolic power and perceived muscle soreness [15,16]. Other research in men’s rugby and soccer has focused on using acceleration and deceleration zones from GNSS data as a unique approach to characterizing objective load metrics [3,12,13,15,16].

### 1.3. Athlete Workload in Rugby Sevens–Considerations for Acceleration as Workload

Despite showing great potential as an approach to characterizing athlete load, existing methods for evaluating the impact of accelerations and decelerations on athlete load vary substantially and often lack precision. While either representing acceleration as all positive values (Delaney et al., 2016), quotients (Delaney et al., 2018), or fixed thresholds (Furlan et al., 2015), all approaches potentially oversimplify the distinct effects or acceleration and deceleration, which may limit the unique and important impact of acceleration and deceleration on athlete load [15,16,17]. Furthermore, while this novel process for evaluating workload may be applied similarly across both male and female athletes, work by Clarke et al. (2017) and Mara et al. (2017) suggest that the use of zone-based data must be appropriately adjusted to the team, as the proprietary zones used in commercial GNSS units or recommended in the literature, are not suitable for use in female team sport populations [18,19].

Before considering the use of acceleration and deceleration derived from GNSS sensor data to characterize athlete load in team sports, it is important to appreciate how they are distinctly developed. Acceleration is normally associated with a volitional increase in speed to overcome an opponent or gain a positional advantage [20]. Deceleration may result from volitional slowing to evade contact or may be necessary during a defensive event. Deceleration may also result from contact with an opponent. Given that changes in direction are common in invasion sports like rugby, there is great potential for acceleration and deceleration events to separately and uniquely contribute to athlete load [21]. Therefore, acceleration and deceleration events may need to be quantified separately as they may be related to different aspects of effort as well as related to different physical mechanisms [21].

### 1.4. Athlete Workload in Women’s Rugby Sevens–Combining Multiple Objective Factors

While there have been important research findings in men’s rugby concerning the characterization of objective load metrics through GNSS sensor data, more recent work suggests that ways of identifying load in the men’s game do not appropriately quantify the experience of female players, with speculation pointing to the generally lower masses of female players as a contributing factor [22]. With the growth of women’s rugby internationally following the inclusion of women’s sevens in the Olympic Programme, it is essential that salient factors of competition load relevant for women’s sevens athletes are quantified [23]. As there is substantial evidence that multiple interacting factors contribute to athlete load in team sports, there is a great need to develop statistical approaches that can fuse data from multiple sources and domains. Furthermore, the specific inclusion of acceleration and deceleration zones alongside other salient characteristics, such as contacts, has not yet been investigated as an approach to quantify drivers of athlete load in women’s rugby sevens. To support this, a linear mixed effect model (LMM) presents a powerful, repeatable, and reliable statistical approach that can help fuse GNSS sensor data with other categorical variables, such as athlete- or match-specific characteristics, to improve the prediction of load, as successfully demonstrated by Iannaccone et al. (2021) [24]. By including multiple factors, LMMs can better model the relationship between training load and performance while considering individual variability across athletes, leading to more accurate insights into their training adaptations and injury risk. The identification of the main factors of load relevant for women’s sevens athletes may enable a simple, targeted, and actionable approach to load monitoring and management compared to the conventional strategies of a sRPE measure or the use of proprietary approaches by commercial software.

Therefore, this investigation’s purpose is to present a novel approach to categorize GNSS sensor data based on accelerations across speed zones and combine these data with contact count and athlete mass using LMM to determine the contribution of these variables, alone and together, to the experience of athlete load in women’s rugby sevens. Further, the new model will be compared to the common sRPE approach in an application of athlete load evaluation across games in multiple tournaments. Ultimately, the aim of the investigation is to provide sports coaches and practitioners with key objective metrics that can be developed and applied both in the monitoring of athlete performance in competition, as well as in training to inform decision-making related to tactical and technical parameters that may be impacted by athlete load and better prepare athletes to meet the demands of their sport.

## 2. Materials and Methods

### 2.1. Study Design

This cohort observational study analyzed the contributions of objective measures of physical performance from GNSS sensor kinematics, contact count, athlete mass to athlete workload and sRPE, through a retrospective analysis of data from international women’s rugby sevens competition. The data collected included GNSS-coded measures, coded match footage, athlete self-reported RPE, athlete mass, and match-related data for one team participating in 99 international women’s sevens matches (2017–2020). Matches were played in outdoor uncovered facilities. The relationship between independent variables (athlete mass, movement distance, movement zone, acceleration type by movement zone, and contact count) and the dependent variable (sRPE, as determined by multiplying RPE and playing time), was assessed using a linear mixed model regression.

### 2.2. Subject Information

Nineteen international women’s sevens players in a full-time rugby sevens training programme participated (26.5 ± 4.20 years, 169.5 ± 5.90 cm, 69.7 ± 6.24 kg). Participants volunteered for the study and gave their written informed consent to participate. Ethical approval for the study was obtained from the University’s Human Research Ethics Board, and followed the principles described in the *Declaration of Helsinki*. All data used by researchers were anonymized by team staff prior to analysis. One team of players was included in the study due to limitations in access to additional teams. While this restricted the overall sample size, it enabled a focused analysis within the available population. Consequently, findings should be interpreted with consideration for the specific characteristics of the team studied.

### 2.3. Methodology

During all matches, athletes wore GNSS monitors (Apex v2.50, StatSports, Newry, UK). The monitors were installed between the shoulder blades of the athlete, fixed under the uniform, in a custom vest (Figure 1).

These sensors measured distance in metres, speed in metres per second and playing time in minutes at 10 Hz. Athlete acceleration was calculated as the first derivative of the GNSS speed measurement. The use of GNSS measures of distances and speeds have been demonstrated to be appropriately accurate at resolutions of 5–10 Hz [25,26]. Following each match, data from the GNSS monitors were downloaded from the unit and exported as a CSV. Measures of distances travelled at certain speeds, and accelerations/decelerations were organized into twelve zones (three speed-zones by four acceleration/deceleration zones) to enable the differential contribution of movement speed and acceleration/deceleration zones within the statistical analysis (Figure 2). Preliminary data processing was completed using Python (Python Language Reference, version 2.7, Amsterdam, The Netherlands).

The use of these zones is consistent with previous studies in team sports linking movement speed, and acceleration/deceleration zones to performance demands from soccer, field hockey, and men’s rugby union fifteens and sevens [17,27,28]. Distance travelled, in metres, was divided into three speed-zones: low speeds (walking—0–1.5 m/s), moderate speeds (running—1.5 m/s to each athlete’s entry to sprinting threshold), and high speeds (sprinting-entry to sprinting and above), expressed relative to each athlete to ensure appropriate reflection of the individual efforts of each athlete [29,30]. The use of three zones represents an efficient combination of the traditional five-zone division available in team sport GNSS systems, intended to offer an applied workflow where movements are easy to identify, and communication of results to coaches is efficient in plain language [31,32,33]. Furthermore, each athlete’s entry to the sprinting threshold was individualized based on the athlete’s velocity achieved from 0 to 10 m of a maximal effort 40 m sprint performed in training and updated regularly throughout the season to ensure accuracy over periods of physical development [29,34].

Once categorized in speed zones, an acceleration threshold of 2 m/s^2^ was used to further categorize distances covered at low (<2.0 m/s^2^) and high (>2.0 m/s^2^) accelerations as well as low (<−2.0 m/s^2^) and high (>−2.0 m/s^2^) decelerations [19]. While various means of quantifying acceleration has been presented in the literature, it remains paramount that the use of a threshold is appropriately reflective of the population and as such, 2 m/s^2^ was used to reflect the effort of female team sport athletes as identified by Mara et al. (2017) [15,16,19,22].

A summed count of contacts per player was produced from match footage, evaluated using Sportscode (v11, Hudl, Lincoln, NE, USA).

Mass, in kilogrammes, was collected pre-match using a portable weigh scale (ES-310, Anyload, Burnaby, BC, Canada).

One RPE value per match was self-reported by athletes using a 0–10 scale, familiar to participants from regular use in training and competition, following each match (roughly 30 min after each match) [4]. With all data incorporated, and cases of missingness dropped ahead of analysis, a total of 1002 complete datasets were available for analysis.

### 2.4. Statistical Analysis

An LMM was used to determine the influence of both fixed and random effects on sRPE by athletes. Fixed effects included continuous kinematic variables derived from GPS data, specifically speed zone distance and acceleration/deceleration zone distance. Random effects accounted for inter-individual variability, including categorical variables such as athlete mass and contact count. This modelling approach allowed the integration of continuous and categorical data to predict sRPE, accommodating both overall patterns and individual-specific responses. The LMM was fitted using R statistical software (version 4.2.1, Vienna, Austria).

### 2.5. Supervised Evaluation of LMM

As a means to evaluate the practical applications of the LMM-predicted workloads, the dataset was used to assess the association between workload and game number within a tournament. Rugby sevens tournaments take place over two or three consecutive days, with teams playing five or six games [35]. One repeated-measures ANOVA was performed, to evaluate the association between the sRPE values (actual or LMM-predicted) and the game number within a tournament by the player using R statistical software (version 4.2.1, Vienna, Austria).

## 3. Results

### 3.1. General Summary of Data

On average, athletes experienced 4 ± 2.5 contacts, played for 11.8 ± 4.53 total minutes, experienced an RPE of 7 ± 1.8 au, and subsequently an sRPE of 99.2 ± 44.09 au. The distances covered with standard deviations in each of the speed zones are detailed in Table 1. Sample sensor data providing insight into the speeds and accelerations of players participating in one match are detailed in Figure 3.

### 3.2. Results of Linear Mixed Model

The model results showed a significant main effect of mass, t(18) = −5.27, *p* < 0.01, distance travelled, t(18) = 5.54, *p* < 0.01, contact count, t(18) = 57.05, *p* < 0.01, and low decelerations, t(18) = −4.60, *p* < 0.01. Furthermore, significant interaction effects were demonstrated between low deceleration and low speed, t(18) = 3.05, *p* < 0.01, and low deceleration and high speed, t(18) = 3.78, *p* < 0.01. However, there were no significant main effects for accelerations at any speed, *p* > 0.01.

Overall, 48.7% of the variability of sRPE was accounted for by variables in the model, R^2^adjusted = 0.487. The relationship between sRPE and mass, distance travelled, movement speed, acceleration type, and contact count, with parameter estimates by zone, is further illustrated in Figure 4:

### 3.3. Results of Supervised Evaluation of LMM Predicated sRPE Values

The repeated measures ANOVA demonstrated a significant main effect for game number, F(5, 1974) = 11.00, *p* < 0.01. There was not a significant main effect found for sRPE type, F(1, 1974) = 0.004, *p* = 0.951 (Figure 5).

A post hoc Tukey test identified significant differences between game number 1 and game numbers 3, 4, 5, and 6, as well as game number 2 and game numbers 3, 4, 5, and 6.

There was also a significant interaction effect between game number and sRPE type, F(5, 1974) = 8.41, *p* < 0.01, accounting for within-player effects (Figure 6). Pairwise comparisons were performed to assess the differences between the sRPE type (actual and predicted) by game number (1–6), and significant differences were identified between the sRPE type at games number 1, 2, 3, and 5 (*p* < 0.01).

## 4. Discussion

### 4.1. General Findings

This investigation has demonstrated that objective factors, categorized by accelerations and decelerations derived from athlete-worn GNSS sensor data across different speed zones, as well as a quantification of contacts and inclusion of athlete mass, can provide a reasonable prediction of athletes’ perceived workload in women’s rugby sevens. The use of different categories of acceleration/deceleration and speed zones presents a novel approach to quantifying the contribution of athlete movement to athlete load using GNSS sensors. Additionally, the significant contribution of distance travelled during high decelerations at low and moderate speeds to athlete load in this cohort provides the impetus for future research. The significant effect of contact affirms the unique role of physical contact in athlete load in the sport of rugby and, therefore, should be accounted for in female rugby athlete load monitoring [8]. Furthermore, the significant effect of mass supports the important consideration of including a range of masses in the development of predictive load algorithms [18].

### 4.2. Distance at Low and Moderate Speed as a Driver of Athlete Load

A novel finding in this study is the significant contribution of distance travelled under high decelerations in low and moderate speed zones on the prediction of athletes’ perceived load. The model demonstrated that accelerations (positive) did not have a significant effect on athletes’ perceived load. This is contrary to research in other field sports, notably soccer, where acceleration is a key metric used in forecasting load [19,34]. This difference in findings concerning the contribution of acceleration/deceleration in athlete load may be due to the sanctioned physical contact in rugby relative to other field sports or potentially due to methodological considerations. In rugby, as opposed to other field sports, deceleration may come from the athlete volitionally slowing to step around another player. Alternatively, it may come from the athlete encountering another body in physical contact, from a tackle, carry, or fend [36]. However, in this model, the unique causes of deceleration (negative acceleration) were not isolated, and as such, physical contacts may be captured in the deceleration categories as well as in the contact count. Further investigation into what contributes to the nature of deceleration in each category is essential to appropriately quantify the potential role of contact on athlete deceleration. For example, as a potential interpretation of the significance of high deceleration at low speeds (Figure 2), it is possible that when scrummaging or rucking substantive force to cause a high deceleration occurrence as bodies come in contact at close distances. Alternatively, players on the edges of the field, running at moderate speed, may experience a high deceleration as they react to play, cutting to evade an opponent or sharply changing direction in response to adjustments of the defensive line. As cases of high deceleration across different movement, speed remains a strong influencing factor of sRPE in women’s rugby sevens; more research investigating the exact nature of these relationships is needed.

### 4.3. Deceleration as a Driver of Athlete Load

A further consideration for evaluating the significant contribution of deceleration to athlete load stems from methodological considerations. For example, Delaney et al. (2016) chose to express all acceleration and deceleration values as positive values to give an indication of total accelerations [16]. Alternatively, Delaney et al. (2018) divided accelerations by decelerations, resulting in a quotient value for an analysis [15]. Similarly, Furlan et al. (2015) used thresholds of acceleration and deceleration to explore the metabolic demands of men’s sevens [17]. The present study categorized acceleration and deceleration using one threshold, applied both as a positive and negative value to create four possible outcomes, which meant there were more opportunities for some element of deceleration (high or low) to influence load in the model. The considerations for acceleration information are further summarized in Table 2.

The current model demonstrates that decelerations do have a significant role in athlete load in women’s sevens; however, the variety of methodological approaches to handling acceleration and deceleration creates discrepancies in reporting [13]. This makes the role of deceleration unclear in the literature as high deceleration cases may be included within a low acceleration threshold as deceleration is a negative value or may be processed to become a positive value [12,13]. Nevertheless, deceleration is present in field sports and, as such, is warranted to be included in training load models [13,15,16,31]. Ultimately, clarity in methodological constructs will support further development of training load models as our study suggests that high deceleration cases are particularly impactful on sRPE.

### 4.4. Physical Contact as a Driver of Athlete Load

The inclusion of particular, unique, sport-specific features in monitoring athlete load has been identified as a meaningful way of producing actionable data that can be used to positively influence athlete performance [10]. In this case, planned physical contact between athletes is a unique feature of rugby compared to non-contact field sports like soccer, where collisions between athletes are incidental. Roe et al. (2017) demonstrated that in men’s rugby union training, sessions with contact resulted in different sRPE than training sessions without contact [37]. Tierney et al. (2021) found differences in collisions, count of contacts and GNSS-software quantified intensity by level of competition [38]. This, coupled with the work of Clarke et al. (2017), which points to sex-based differences in contact loading, suggests that collisions warrant inclusion in athlete load models across rugby codes (league, union, sevens) as well as across sexes [18].

### 4.5. Athlete Mass as a Driver of Athlete Load

Interestingly, athlete mass was a significant main effect in the model, suggesting that mass influences sRPE in this cohort. This finding is in alignment with studies examining male rugby players, suggesting that mass influences key performance metrics around contact, speed, running distances, and strength, all factors that may be tied to athletic expression in the various rugby codes [4,36]. However, in this particular study, the mass had a significant negative effect, whereby athletes with higher masses experienced lower sRPE loads compared to athletes with lower masses. This may be due to tactical strategies employed in the game whereby heavier players may be predisposed to carrying the ball into contact and/or making key tackles as a part of the defensive line, a common strategy in fifteens [36]. Notably, previous research linking mass to rugby performance comes from samples of rugby fifteens athletes. Rugby sevens generally tend to have a more homogenous population in terms of mass for both male and female players relative to fifteens, with players sharing similar tactical and physical demands when the ball is in play; however, sevens retains the two main positional differences in forwards and backs [39]. Forwards generally have higher masses than backs, so it is possible that these players are experiencing unique positional demands, as suggested by Misseldine et al. (2018) [40]. While the present study used data from a women’s sevens population, further investigations across rugby codes for both male and female athletes should consider the roles of decelerations, contact, and mass in athlete load.

### 4.6. Application of Supervised Example

The results of the supervised example comparing LMM predicted and actual sRPE identified that each sRPE type demonstrated the same significant differences between games. Also, the lack of statistically significant differences between predicted and actual sRPE values, suggest that the LMM produces a workload comparable to that of the traditional sRPE calculations. However, there was a significant interaction between sRPE types, and game and pairwise comparisons demonstrated that predicted values were significantly higher than actual values for the first two games, and significantly lower than the actual values for game numbers three to six. Practically, this infers that when using either model in an applied sense, coaches will be able to draw the same conclusions regarding relative differences between games, yet there may be additional details to be investigated surrounding the use of either the traditional or LMM techniques with respect to bias and magnitude.

Overall, this suggests that, in this example, either model is internally consistent but may not be able to be used in an integrated sense. From the point of interpretation, regardless of approach, traditional or LMM-predicted, the findings showed that the first two games of the tournaments had a significantly lower load than the last four games. This is similar to a phenomenon observed in men’s basketball whereby sRPE increased in a playoff versus regular (pool) play and in final matches compared to semifinals, suggesting that athletes’ perceptions surrounding the importance of the match may affect their reported RPE and, subsequently, their sRPE [41,42]. However, the greater variability in the actual sRPE than the predicted (Figure 6), could also suggest that increases in workload may be explained by psychological aspects of the competition environment, such as an identified value of playing in a particular match or against a particular opponent, both of which elicit the perception of increased effort [42]. This may point to a limitation of using objective load measurement whereby psychological perception may not be accounted for.

The deployment of any strategy requires balancing resource and performance demands. The collection of athlete-reported measures has improved using athlete monitoring apps, telecommunications, and survey tools. However, it remains the responsibility of both athletes and practitioners to ensure the completeness of these datasets [43]. This can be particularly challenging in competition settings, where athletes experience demands outside their usual routines, such as media obligations, anti-doping protocols, medical treatments, and the presence of spectators [44]. Additionally, in some professional sports, athletes and their unions have raised concerns about the collection of self-reported data, citing issues around athlete autonomy and the potential misuse of data in decisions regarding playing time [45].

### 4.7. Future Considerations

The use of GNSS sensor data to obtain distances and speeds, as well as to calculate accelerations and decelerations, presents a simple, efficient strategy for data collection [26]. It is worth noting that multisensory approaches leveraging inertial measurement units (IMUs) for the collection of speed and acceleration data are also popular strategies in athlete monitoring [25]. However, the processing strategies to fuse data between GNSS and IMU sensors in commercially available products are not well known [26]. Time intervals, smoothing, and filtering strategies are not always publicly available to practitioners, and as such the use of GNSS data offers data that is sufficiently robust and transparent [26,32]. Further investigation into sensor fusion methodologies present in commercially available products, as well as the validity across sporting populations, is encouraged.

### 4.8. Practical Applications for Coaches and Practitioners

This investigation identified the contribution of several meaningful variables to athlete load in women’s rugby sevens, with implications for both the competition and training environments.

High deceleration events are particularly impactful to athlete load in women’s sevens rugby. In competition, monitoring these events through post-game analysis of GNSS data may help coaches gain insight into player experiences from the previous match and encourage adjustments to line-ups and substitution plans for subsequent matches. In training, practitioners should focus on training for optimal movement strategies, whether through self-slowing and cutting drills or in rugby contact drills around tackle or carry technique, to ensure athletes are able to safely experience the demands of high deceleration [46,47].

Practitioners are encouraged to consider the inclusion of physical contact in their load monitoring for contact sports. In the competition environment, the monitoring of contact events may also support team strategies around managing player load across a tournament, such as when to rest certain players. The sevens tournament format is such that teams of twelve athletes will play five or six games in a two- or three- day period, making roster management a critical element of team success [17,35]. In training, a count of contacts in drills may offer additional insight into athlete loading and further support periodization of load to ensure both athlete safety and appropriate skill development, particularly around contact skills [48,49].

Finally, practitioners are strongly encouraged to determine relevant drivers of athlete load for their sport environment and clearly document how their data are measured to ensure high-quality, reproducible cases.

## 5. Conclusions

This investigation presents a novel approach to the quantification of athlete movement as a contributor to athlete load in competition. The model explained 48.7% of the global variance of sRPE, with high decelerations at low and moderate speeds, athlete mass, and contact count, proving significant factors. The significant contribution of distance in high decelerations at low and moderate speeds invites practitioners to consider the role of deceleration in women’s rugby sevens as an important load metric. Additionally, the significant contribution of mass and contact continues to attest to the importance of considering sport-specific factors in athlete load monitoring. Taken together, the results infer that objective metrics like decelerations by speed zone, athlete mass, and contact count may be useful in the quantification of athlete load in women’s rugby sevens.

## Figures and Tables

**Figure 1 sensors-24-06699-f001:**
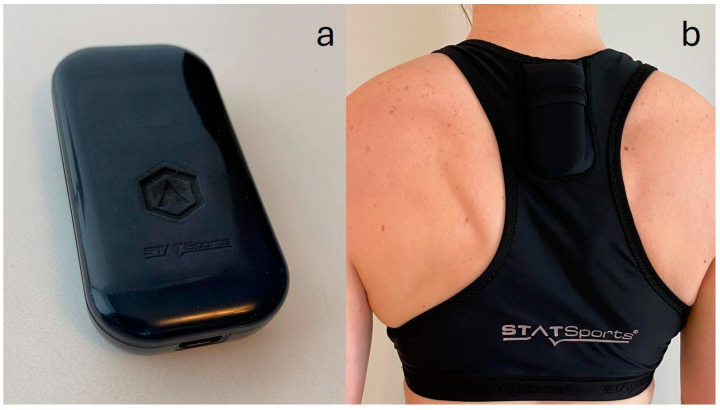
(**a**) GNSS monitor used in data collection and (**b**) GNSS monitor in vest worn by athlete.

**Figure 2 sensors-24-06699-f002:**
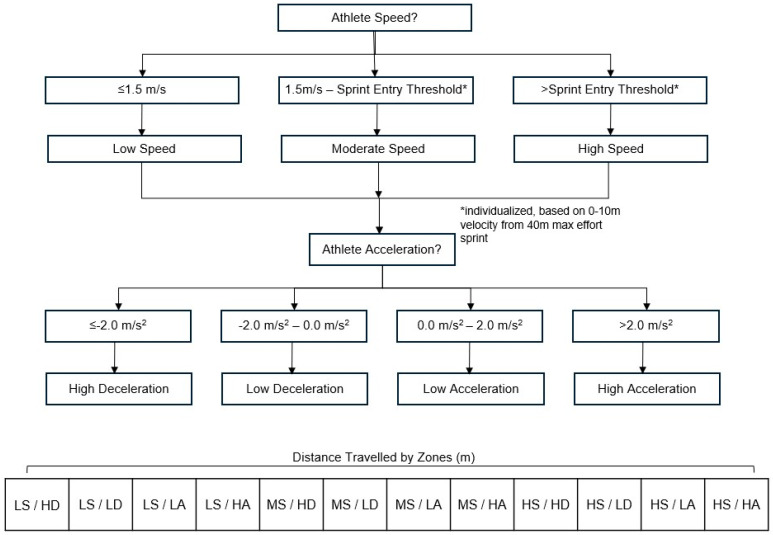
Visual depiction of organization of movement categories based on low, moderate, and high athlete speeds and low or high accelerations or decelerations, resulting in 12 zones.

**Figure 3 sensors-24-06699-f003:**
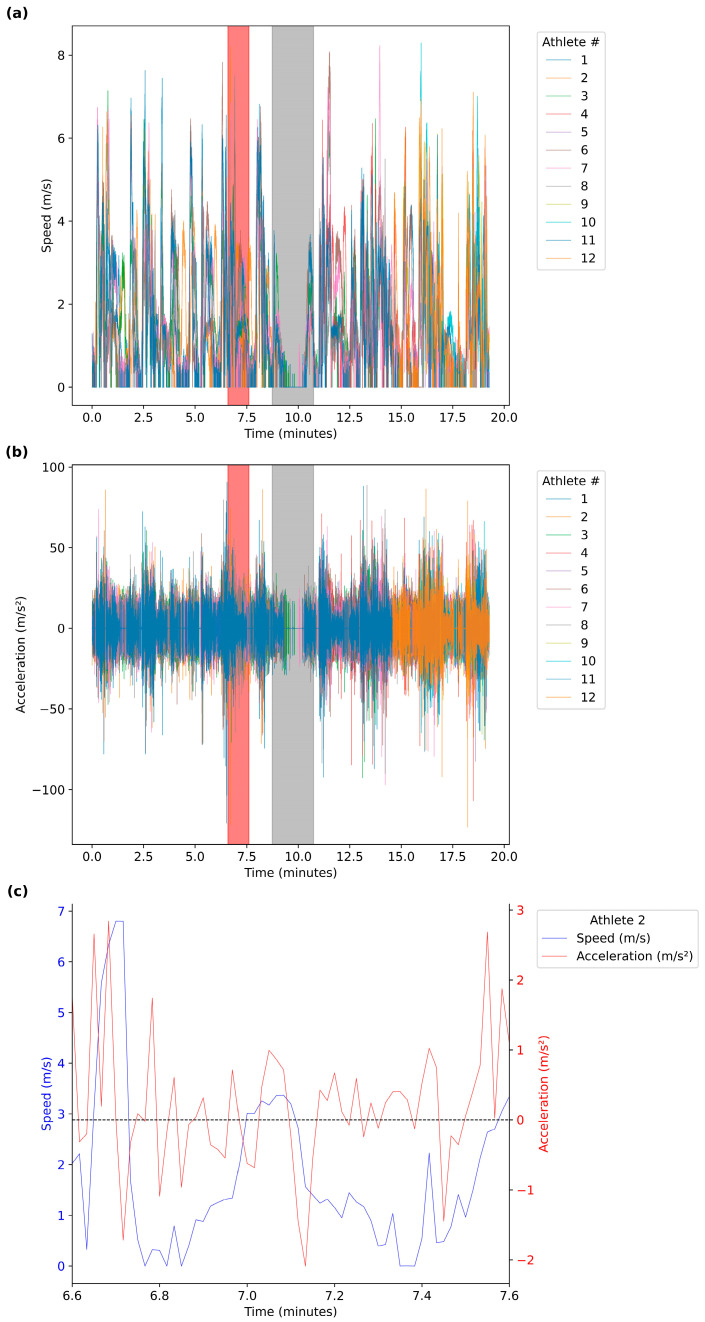
Sample sensor data from one match showing: (**a**) all athlete speeds, (**b**) all athlete accelerations and decelerations, and (**c**) a single athlete subset for a one minute period of play. The grey shaded box in (**a**,**b**) denotes the two minute halftime period of the match, where players remain on field. The red shaded box in (**a**,**b**) denotes the subset time period in (**c**). The dashed line in (**c**) denotes a zero acceleration, above this line are positive accelerations and below are negative accelerations or decelerations.

**Figure 4 sensors-24-06699-f004:**
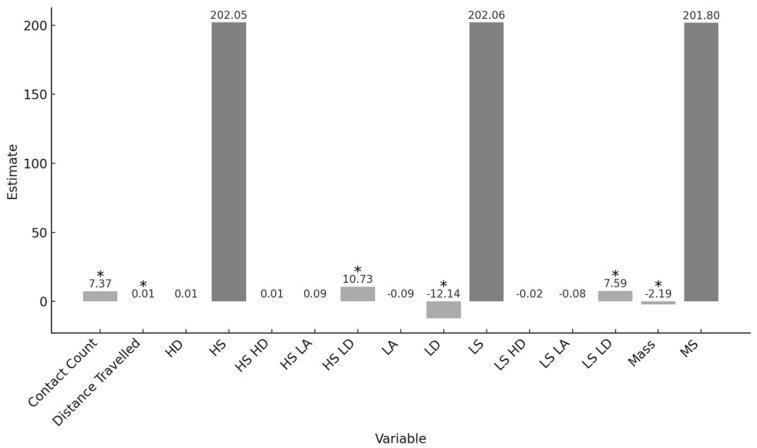
Parameter estimates by variable, including zones, * denotes statistical significance (*p* < 0.01).

**Figure 5 sensors-24-06699-f005:**
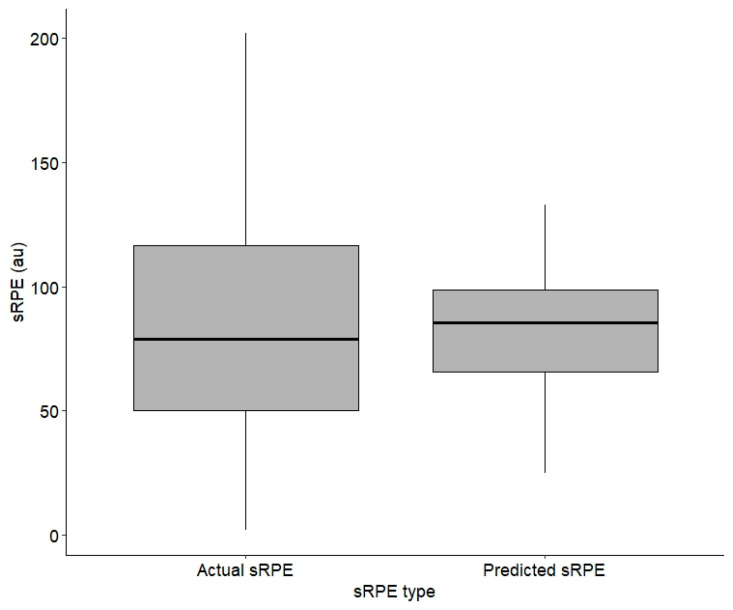
sRPE type (actual or predicted) by sRPE value from supervised example.

**Figure 6 sensors-24-06699-f006:**
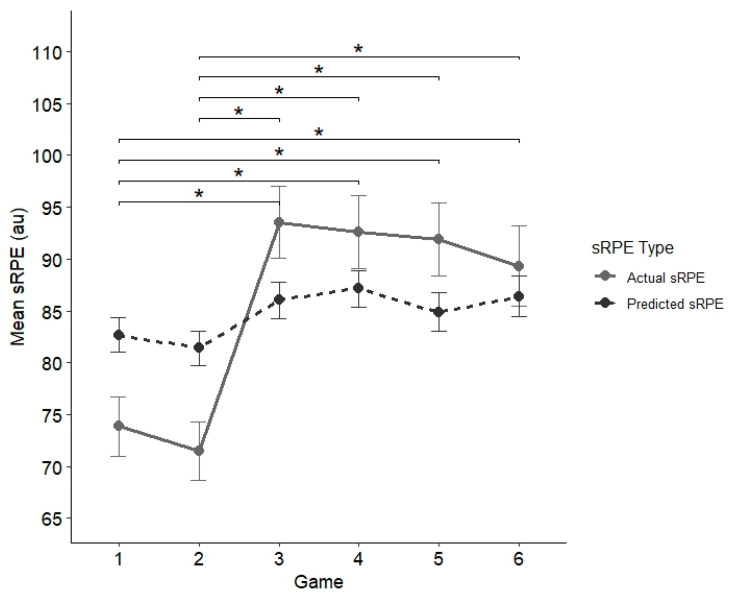
Interaction effect of sRPE type and game number on sRPE value, * denotes statistical significance (*p* > 0.01) within sRPE type, not between sRPE type.

**Table 1 sensors-24-06699-t001:** Distances covered by category of movement speed and acceleration type, including standard deviations.

Movement Speed	Acceleration Type	Distance Covered (m)
Low Speed (LS)	Low Acceleration (LA)	11.24 ± 5.655
	High Acceleration (HA)	1.99 ± 1.300
	Low Deceleration (LD)	259.39 ± 130.090
	High Deceleration (HD)	2.85 ± 9.401
Moderate Speed (MS)	Low Acceleration (LA)	21.67 ± 9.616
	High Acceleration (HA)	16.82 ± 7.385
	Low Deceleration (LD)	703.58 ± 298.059
	High Deceleration (HD)	16.41 ± 6.993
High Speed (HS)	Low Acceleration (LA)	2.37 ± 1.603
	High Acceleration (HA)	2.60 ± 1.636
	Low Deceleration (LD)	82.49 ± 52.290
	High Deceleration (HD)	1.78 ± 1.314

**Table 2 sensors-24-06699-t002:** Methodological approaches to processing acceleration and deceleration data.

Authors	Population	Strategy for Data Processing	Data Processing Output
Delaney et al. (2016) [16]	Men’s rugby league	Absolute value (all values expressed as positive acceleration)	Total acceleration distance
Delaney et al. (2018) [15]	Men’s rugby league	Accelerations/Decelerations	Acceleration ratio
Furlan et al. (2015) [17]	Men’s rugby sevens	Thresholds of acceleration and deceleration	Four bins of acceleration and deceleration distances
Current study	Women’s rugby sevens	Thresholds of speed and acceleration and deceleration	Twelve bins of acceleration and deceleration distances

## Data Availability

The data presented in this study are available on request from the corresponding author, A.E.S., due to the sensitive nature of high performance sport data.

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
