# Peer review of "Predicting Athlete Workload in Women’s Rugby Sevens Using GNSS Sensor Data, Contact Count and Mass"

_sensors, 2024, doi:10.3390/s24206699_

Round 1

Reviewer 1 Report

Comments and Suggestions for Authors

The manuscript proposed an athlete workload predicting method in women’s rugby sevens using multi-source data. The process is innovative and insightful, reflecting the authors’ profound professional skills. However, before this article is published, I think several key issues should be resolved first.

1.      This manuscript provides a clear description of the existing problems of current athlete load prediction methods, and conducts in-depth thinking and exploration based on the issues. However, the calculation method used in the study is still vague. It would be better to provide a detailed calculation process, especially for the fusion method of multi-source data. Could the linear mixed effect model used in the experiment be explained and analyzed in detail?

2.      The article obtained sufficient experimental data, but the experimental analysis process was designed to be a bit too simple. For example, the prediction process of the athlete's workload did not seem to be considered in the experiment. There was also a lack of comparison with other sports and comparison of male and female athletes, which was not enough to support the conclusions drawn through subsequent discussions in the article. It is recommended to enrich and improve the experimental process and results.

3.         In the last section of the article, a lot of discussions were carried out and many scholars' previous explorations were cited. I think the background, current status, and problems of existing methods can be placed in the first section to highlight the importance of this study.

Comments on the Quality of English Language

1. In Line 222, "... ... positively influence athlete performance5. "

    The number 5 may be a reference. 

2. In Line 232, "This is finding is in alignment from studies with ... ..."

    There are two "is" in this sentence.

Author Response

Comments and Suggestions for Authors

The manuscript proposed an athlete workload predicting method in women’s rugby sevens using multi-source data. The process is innovative and insightful, reflecting the authors’ profound professional skills. However, before this article is published, I think several key issues should be resolved first.

  1. This manuscript provides a clear description of the existing problems of current athlete load prediction methods, and conducts in-depth thinking and exploration based on the issues. However, the calculation method used in the study is still vague. It would be better to provide a detailed calculation process, especially for the fusion method of multi-source data. Could the linear mixed effect model used in the experiment be explained and analyzed in detail?

We thank the reviewer for this question. To provide more detail on the linear mixed effect model and its utility we have added text in both the introduction (Lines 93-102) as well as in the materials and methods section (Lines 176-184).

  1. The article obtained sufficient experimental data, but the experimental analysis process was designed to be a bit too simple. For example, the prediction process of the athlete's workload did not seem to be considered in the experiment. There was also a lack of comparison with other sports and comparison of male and female athletes, which was not enough to support the conclusions drawn through subsequent discussions in the article. It is recommended to enrich and improve the experimental process and results.

We thank the reviewer for this comment. We have attempted to enhance the information around the prediction process through inclusion of additional information on the use of the linear mixed model as a predictor of athlete load in the introduction (Lines 93-102) as well as in the materials and methods section (Lines 176-184). Bartlett et al. (2021) identify the importance of simplicity in ease of use in implementing any data collection and knowledge translation strategy in an applied sports environment. We believe that the mixed model approach allows practitioners to use simple data from GNSS and athlete/match characteristics to accurately predict subjective load, ensuring both a statistically robust approach but one that is accessible and efficient to applied sport practitioners (Lines 37-38, 93-106). We have also included a particular conclusions section to support further clarity of the experimental process and results (Lines 317-327). Finally, we also recognize the lack of comparison between male and female athletes and across sports as a limitation of this paper. While we do not have data for other populations, we have highlighted this as a limitation of the current analysis in the discussion (Lines 304-306; 314-316).

Reference for response:

Bartlett JD, Drust B. A framework for effective knowledge translation and performance delivery of Sport Scientists in professional sport. European Journal of Sport Science. 2021;21(11). p. 1579-f1587. doi:10.1080/17461391.2020.1842511  

  1. In the last section of the article, a lot of discussions were carried out and many scholars' previous explorations were cited. I think the background, current status, and problems of existing methods can be placed in the first section to highlight the importance of this study.

We thank the reviewer for this comment. We have attempted to provide additional context to the data in a variety of ways. In the introduction we have tried to better represent the current status and problems of existing methods by more clearly stating these concepts in the introduction (Lines 59-71) as well as summarizing some of the points that are presented in more detail in the discussion (Lines 248-267).

Sample sensor data is provided in a new figure (Figure 2 – about Line 191). Additional details around the sensor placement and post-match data collection were provided on Line 117-118 and Lines 129-130. Further detail on data processing tools were added on Lines 139-140. Enhanced sections in the introduction bolster key considerations for evaluating workload (Lines 59-71; 93-106). An adjusted discussion section includes an improved section on data handling (Lines 248-267) now, summarized in a table (new Table 2 – about Line 58) as well as an additional section on considerations for multi-sensor fusion (Lines 307-316). Finally, the inclusion of a specific conclusion section supports enhancements to both the content and quality of the paper.

Comments on the Quality of English Language

  1. In Line 222, "... ... positively influence athlete performance5. "

    The number 5 may be a reference. 

We thank the reviewer for the observation and have adjusted the format to reflect that this was meant as a reference (Line 279).

  1. In Line 232, "This is finding is in alignment from studies with ... ..."

    There are two "is" in this sentence.

We thank the reviewer for this edit and have adjusted the sentence for clarity (Line 289).

Reviewer 2 Report

Comments and Suggestions for Authors

This work presents an approach to the quantification of athlete movement as a contributor to athlete load in women’s rugby sevens by using GNSS sensor. This manuscript needs to address the following issues before it can be considered for publishing.

1.     What is the novelty of the proposed method compared with the conventional sports evaluation method?

2.     The full name of a GNSS sensor should be pointed out.

3.     What is the difference in evaluating workload between men’s and women’s rugby sevens?

4.     There are only two figures in the paper, which is insufficient to support the viewpoint, more sports data recorded by the sensor should be provided.

Author Response

This work presents an approach to the quantification of athlete movement as a contributor to athlete load in women’s rugby sevens by using GNSS sensor. This manuscript needs to address the following issues before it can be considered for publishing.

  1. What is the novelty of the proposed method compared with the conventional sports evaluation method?

We thank the reviewer for this question and have added further detail to the introduction to contextualize the novelty of our proposed method. In lines 93-102 we expand on the importance of having objective variables as a means to have both improved understanding and actionable outcomes to use to monitor and manage athlete load relative to the conventional strategy of a holistic subjective load measure.

  1. The full name of a GNSS sensor should be pointed out.

We thank the reviewer for this observation and have added the full name for global navigation satellite system (GNSS) sensors in both the abstract (Lines 12-13) and introduction (Line 45).

  1. What is the difference in evaluating workload between men’s and women’s rugby sevens?

We thank the reviewer for this question and have added further information to the introduction to highlight considerations and differences in evaluating workload between men’s and women’s rugby sevens players. In lines 66-71, we further build on the processes outlined, adding that the work of Clarke et al. (2017) and Mara et al. (2017) specifically with female populations have found that zone-based data requires additional consideration and adjustment and that the proprietary zone settings available in commercial units are not appropriate for female team-sport athletes.

References for Response:

Clarke AC, Anson JM, Pyne DB. Proof of Concept of Automated Collision Detection Technology in Rugby Sevens. Journal of Strength and Conditioning Research. 2017;31(4). p. 1116-1120. doi:https://doi.org/10.1519/jsc.0000000000001576

Mara JK, Thompson KG, Pumpa KL, Morgan S. The acceleration and deceleration profiles of elite female soccer players during competitive matches. Journal of Science and Medicine in Sport. 2017;20(9). p. 867-872. doi:10.1016/j.jsams.2016.12.078

  1. There are only two figures in the paper, which is insufficient to support the viewpoint, more sports data recorded by the sensor should be provided.

We appreciate this insight from the reviewer; we have adjusted the formatting of Table 1 and further amended the formatting of the figures to reflect a higher quality of work (updated Figure 1 – about Line 141; updated Table 1 – about Line 190; new Figure 2 – about line 191; updated Figure 3 – about line 207; new Table 2 – about Line 258). Further, we have specifically included additional sample data in the new Figure 2 (about Line 191) which reflects athlete speed and acceleration data from one match to provide further context to the reader.

Reviewer 3 Report

Comments and Suggestions for Authors

This study delves into how mass, contact, and accelerations and decelerations at varying speeds impact the load in women's rugby sevens. The research is indeed intriguing, yet I believe it necessitates substantial supplementation and revisions to meet publication standards.

  1. What conclusions have been drawn from the analysis, data, and discussions presented in this paper?

  2. Regarding the formatting, Figure 1 appears too small, whereas Table 1 is excessively large. The quality of the figures and tables in this manuscript falls significantly short of what is commonly seen in articles published in journals like "Sensors."

  3. The paper is quite brief. Even in the format of communications or letters, it seems to lack sufficient content. I recommend supplementing it with a wealth of intermediate results, such as the sensor model used, installation methods, preliminary data processing steps, and corresponding findings.

  4. The study solely focuses on GPS/GNSS. However, it's crucial to note that GPS positioning errors can be significant under semi-indoor conditions due to signal obstruction or venue interference. Multi-sensor fusion approaches are commonly employed in similar research contexts and should be acknowledged and discussed.

  5. The discussion section is quite verbose; could some of the information be presented more concisely and visually through tables, providing a clearer and more intuitive representation?

Comments on the Quality of English Language

Acceptable

Author Response

This study delves into how mass, contact, and accelerations and decelerations at varying speeds impact the load in women's rugby sevens. The research is indeed intriguing, yet I believe it necessitates substantial supplementation and revisions to meet publication standards.

1. What conclusions have been drawn from the analysis, data, and discussions presented in this paper?

We thank the reviewer for this comment, to improve clarity we have added a specific conclusion section, adjusting the discussion to ensure concluding remarks are appropriately placed in the conclusion (Lines 317-327).

2. Regarding the formatting, Figure 1 appears too small, whereas Table 1 is excessively large. The quality of the figures and tables in this manuscript falls significantly short of what is commonly seen in articles published in journals like "Sensors."

We appreciate this insight from the reviewer; we have adjusted the formatting of Table 1 and further amended the formatting of the figures to reflect a higher quality of work (updated Figure 1 – about Line 141; updated Table 1 – about Line 190; new Figure 2 – about line 191; updated Figure 3 – about line 207; new Table 2 – about Line 258). Further, we have specifically included additional sample data in the new Figure 2 (about Line 191) which reflects athlete speed and acceleration data from one match to provide further context to the reader.

3. The paper is quite brief. Even in the format of communications or letters, it seems to lack sufficient content. I recommend supplementing it with a wealth of intermediate results, such as the sensor model used, installation methods, preliminary data processing steps, and corresponding findings.

We appreciate this comment from the reviewer and have attempted to provide additional context to the data in a variety of ways. Sample sensor data is provided in a new figure (Figure 2 – about Line 191). Additional details around the sensor placement and post-match data collection were provided on Line 117-118 and Lines 129-130. Further detail on data processing tools were added on Lines 139-140.

The primary author also personally corresponded with the commercial sensor company to confirm chip set (Ublox) and antenna hardware (Taoglas Embedded). However, at this time, the company was not able to confirm specific model information relating to the chip set and antenna hardware and as such only the sensor model, as identified by the commercial sensor company was included in the paper (Apex v2.50, StatSports, Newry: UK) on Line 128.

4. The study solely focuses on GPS/GNSS. However, it's crucial to note that GPS positioning errors can be significant under semi-indoor conditions due to signal obstruction or venue interference. Multi-sensor fusion approaches are commonly employed in similar research contexts and should be acknowledged and discussed.

We thank the reviewer for this suggestion. To confirm collection conditions, in open-air, outdoor, uncovered playing fields, we added a note on Lines 117-118. We also recognize that the work of Roel et al. (2018) and Malone et al., (2017) suggest that the use of GPS/GNSS sensor data are sufficient for use provided collection is above 5 Hz (Lines 132-134). However, we also recognize that multi-sensor fusion is one approach employed by practitioners working with these sensors, in particular the use of GPS/GNSS and inertial measurement units which are both found in commercially available sports monitoring products and have we considered the importance of this in the discussion section (Lines 307-316).

References for response:

Roell M, Roecker K, Gehring D, Mahler H, Gollhofer A. Player monitoring in indoor team sports: concurrent validity of inertial measurement units to quantify average and peak acceleration values. Frontiers in physiology. 2018;9(141). doi.org/10.3389/fphys.2018.00141

Malone JJ, Lovell R, Varley MC, Coutts AJ. Unpacking the Black Box: Applications and Considerations for Using GPS Devices in Sport. International Journal of Sports Physiology and Performance. 2017;12(Suppl 2):S218-S226. doi: 10.1123/ijspp.2016-0236

5. The discussion section is quite verbose; could some of the information be presented more concisely and visually through tables, providing a clearer and more intuitive representation?

We thank the reviewer for this comment. This encouraged us to return to the discussion and adjust the portion on data handling (Lines 248-267) and instead summarize the information in a table (new Table 2 – about Line 258).

Reviewer 4 Report

Comments and Suggestions for Authors

This paper has very little content(10 pages include References) and only two figure and one table. Figure is very rough and table  is not standardized. Meanwhile, there is no conclusion in this paper. Overall, it does not meet academic paper requirements.

Detailed comments:

Point 1: Line 12, Add the full name about GNSS.

Point 2: Line 22, the keywords of GPS is not in Abstract. Please delete it.

Point 3: Figure 1 and 2 is very rough.

Point 4: Table 1 is not standardized. 2.85 ± 9.401 is unreliable results, error is greater than real distance covered.

Point 5: Line 57, the style of reference seem to wrong.

Point 6: Line 139 and 140, delete row.

Point 7: There is no conclusion in this paper.

Point 8: This paper has very little content and only two figure and one table, as well as 10 pages include Regerences, meanwhile. It does not meet academic paper requirements.

Author Response

This paper has very little content(10 pages include References) and only two figure and one table. Figure is very rough and table  is not standardized. Meanwhile, there is no conclusion in this paper. Overall, it does not meet academic paper requirements.

Detailed comments:

Point 1: Line 12, Add the full name about GNSS.

We thank the reviewer for this observation and have added the full name for global navigation satellite system (GNSS) sensors in both the abstract (Lines 12-13) and introduction (Line 45).

Point 2: Line 22, the keywords of GPS is not in Abstract. Please delete it.

We thank the reviewer for this observation, we have removed GPS as a keyword (Line 22).

Point 3: Figure 1 and 2 is very rough.

We appreciate this insight from the reviewer; we have adjusted the formatting of Table 1 and further amended the formatting of the figures to reflect a higher quality of work (updated Figure 1 – about Line 141; updated Table 1 – about Line 190; new Figure 2 – about line 191; updated Figure 3 – about line 207; new Table 2 – about Line 258). Further, we have specifically included additional sample data in the new Figure 2 (about Line 191) which reflects athlete speed and acceleration data from one match to provide further context to the reader.

Point 4: Table 1 is not standardized. 2.85 ± 9.401 is unreliable results, error is greater than real distance covered.

We thank the reviewer for this comment, we apologize for the lack of clarity as the summary data is meant to describe mean and standard deviations, not standard errors. To remedy this we have adjusted the description of the Table (Line 188) and the title of the Table (about Line 190). In this case, the mean represents the team average where the larger deviations represent the variance between single athletes in terms of distancers covered for those movement categories.

Point 5: Line 57, the style of reference seem to wrong.

We thank the reviewer for this edit and have adjusted the reference format to match the prescribed style (Line 59).

Point 6: Line 139 and 140, delete row.

Thank you to the reviewer for this edit, we have removed the blank row (previously line 139; adjusted range is around line 171).

Point 7: There is no conclusion in this paper.

We thank the reviewer for this comment, to improve clarity we have added a specific conclusion section, adjusting the discussion to ensure concluding remarks are appropriately placed in the conclusion (Lines 317-327).

Point 8: This paper has very little content and only two figure and one table, as well as 10 pages include Regerences, meanwhile. It does not meet academic paper requirements.

We thank the reviewer for this comment. We have attempted to provide additional context to the data in a variety of ways. Sample sensor data is provided in a new figure (Figure 2 – about Line 191). Additional details around the sensor placement and post-match data collection were provided on Line 117-118 and Lines 129-130. Further detail on data processing tools were added on Lines 139-140. Enhanced sections in the introduction bolster key considerations for evaluating workload (Lines 59-71; 93-106). An adjusted discussion section includes an improved section on data handling (Lines 248-267) now, summarized in a table (new Table 2 – about Line 58) as well as an additional section on considerations for multi-sensor fusion (Lines 307-316). Finally, the inclusion of a specific conclusion section supports enhancements to both the content and quality of the paper.

Round 2

Reviewer 1 Report

Comments and Suggestions for Authors

In line 177, "fA linear mixed effects model (LMM) was ..." should be written as "The LMM was ...". There is already an expression in Section 1.

In line 185. "3. Results" should be on a new line.

Author Response

Comments and Suggestions for Authors

In line 177, "fA linear mixed effects model (LMM) was ..." should be written as "The LMM was ...". There is already an expression in Section 1.

We thank the reviewer for this edit and have made the adjustment to lines 232-233.

In line 185. "3. Results" should be on a new line.

We thank the reviewer for this edit and have added a line break (Line 249).

Reviewer 2 Report

Comments and Suggestions for Authors

The authors have address all my questions.

Author Response

The authors have address all my questions.

We thank the reviewer for their insights from the previous round of edits which led to substantial improvements in the quality of the paper.

Reviewer 3 Report

Comments and Suggestions for Authors

I have reviewed your revised article and the response letter. Unfortunately, the quality has not been sufficiently improved. The research is indeed interesting and the work is substantial, and I have no objections to that. Innovative research like this, for instance, the new application of position sensors in the sports field, should be advocated and published. However, the presentation quality of the article remains low. I recommend that you take a look at several other articles published in Sensors, especially how they present figures and tables. Here are my suggestions:

  1. Consider dividing the content into more subsections to avoid long blocks of text spanning an entire page. This will make the article more readable.
  2. Instead of merely fulfilling a requirement, genuinely enhance the quality and quantity of images in your article. High-quality and abundant visuals will make it easier for readers to grasp the content.
Comments on the Quality of English Language

N/A

Author Response

1. Consider dividing the content into more subsections to avoid long blocks of text spanning an entire page. This will make the article more readable.

We thank the reviewer for this suggestion, we have included subheadings throughout the article to more appropriately organize the material and improve readability (Lines 26, 63, 85, 110, 152, 164, 175, 227, 241, 250, 262, 276, 293, 307, 332, 362, 374, 394, 430, 441).

2. Instead of merely fulfilling a requirement, genuinely enhance the quality and quantity of images in your article. High-quality and abundant visuals will make it easier for readers to grasp the content.

We appreciate this feedback and have provided additional visuals (Lines 179, 287, 289) to enhance the methodology. We have also included high-resolution copies of the visuals as an attachment to the overall submission in the event that the materials are being compressed when included in the manuscript document.

We have reviewed other papers published in Sensors as well as recommendations from the Academic Editor and have added an applied example that we believe improves the quality of the manuscript (Lines 241-248,276-291, 394-429).

As a means to ensure an overall high quality of work we have reviewed our manuscript using the STROBE checklist (STrengthening the Reporting of OBservational studies in Epidemiology) to ensure that the manuscript meets reasonable recommendations for cohort observational research. In alignment with the recommendations from this checklist, we have added material to the methods (Line 158, Lines 170-174, Line 225).

Reference for Response

STROBE Initiative. Strobe Statement. 2024. URL http://strobe-statement.org

Reviewer 4 Report

Comments and Suggestions for Authors

I still keep my opinion that this article does not meet academic paper requirements.

Author Response

We appreciate this feedback as it has led us to make some additional changes to enhance the manuscript. We’ve taken a look at other papers published in Sensors as well as recommendations from the Academic Editor and have added an applied example that we believe improves the quality of the manuscript (Lines 241-248,276-291, 394-429). As a means to ensure an overall high quality of work we have reviewed our manuscript using the STROBE checklist (STrengthening the Reporting of OBservational studies in Epidemiology) to ensure that the manuscript meets reasonable recommendations for cohort observational research. In alignment with the recommendations from this checklist, we have added material to the methods (Line 158, Lines 170-174, Line 225).

Reference for Response

STROBE Initiative. Strobe Statement. 2024. URL http://strobe-statement.org